# Preserved wake-dependent cortical excitability dynamics predict cognitive fitness beyond age-related brain alterations

Maxime Van Egroo [1,6], Justinas Narbutas[1,2,6], Daphne Chylinski[1,6], Pamela Villar González[1], Pouya Ghaemmaghami[1], Vincenzo Muto[1], Christina Schmidt[1,2], Giulia Gaggioni[1], Gabriel Besson[1], Xavier Pépin[1], Elif Tezel[1], Davide Marzoli[1], Caroline Le Goff[3], Etienne Cavalier[3], André Luxen[1], Eric Salmon[1,2,4], Pierre Maquet [1,4], Mohamed Ali Bahri[1], Christophe Phillips [1,5], Christine Bastin [1,2], Fabienne Collette[1,2] & Gilles Vandewalle [1]*

Age-related cognitive decline arises from alterations in brain structure as well as in sleep-wake regulation. Here, we investigated whether preserved wake-dependent regulation of cortical function could represent a positive factor for cognitive fitness in aging. We quantified cortical excitability dynamics during prolonged wakefulness as a sensitive marker of age-related alteration in sleep-wake regulation in 60 healthy older individuals (50–69 y; 42 women). Brain structural integrity was assessed with amyloid-beta- and tau-PET, and with MRI. Participants' cognition was investigated using an extensive neuropsychological task battery. We show that individuals with preserved wake-dependent cortical excitability dynamics exhibit better cognitive performance, particularly in the executive domain which is essential to successful cognitive aging. Critically, this association remained significant after accounting for brain structural integrity measures. Preserved dynamics of basic brain function during wakefulness could therefore be essential to cognitive fitness in aging, independently from age-related brain structural modifications that can ultimately lead to dementia.

[1] GIGA-Cyclotron Research Centre-In Vivo Imaging, University of Liège, Liège, Belgium. [2] Psychology and Cognitive Neuroscience Research Unit, University of Liège, Liège, Belgium. [3] Department of Clinical Chemistry, University Hospital of Liège, Liège, Belgium. [4] Department of Neurology, University Hospital of Liège, Liège, Belgium. [5] GIGA-In Silico Medicine, University of Liège, Liège, Belgium. [6] These authors contributed equally: Maxime Van Egroo, Justinas Narbutas, Daphne Chylinski. *email: gilles.vandewalle@uliege.be

Aging is associated with an overall cognitive decline triggered in part by a progressive degradation of brain structure. Limited but notable neuronal loss takes place during healthy adulthood[1]. In addition, tau protein, which stabilizes axonal structure and contributes to synaptic function, and amyloid-beta (Aβ) protein, a peptide directly related to neuronal activity, progressively aggregate in the brain over the lifespan to form neurofibrillary tangles (NFTs) and senile plaques, respectively[2]. Tau NFTs, Aβ plaques, and neurodegeneration favor cognitive decline[3]. They are considered as major underlying causes of dementia and constitute the hallmarks of Alzheimer's disease (AD)[4]. However, age-related changes in brain structure go undetected for decades: tau protein aggregation takes place as early as during the second decade of life in the brainstem, while Aβ aggregates can be detected around the 4th decade in the neocortex[5].

One of the first signs of AD- and age-related brain structural degradation may reside in alterations in the regulation of sleep and wakefulness[6]. Sleep–wake disruption is indeed strongly associated with AD neuropathology[7]: gray matter (GM) integrity has been associated with measures of sleep quality, including sleep slow waves characteristics, in cross-sectional and longitudinal studies[8,9]. Aβ and tau burdens in healthy older individuals have been associated with the amount of slow waves generated during non-rapid eye movement (NREM) sleep[10,11]. Importantly, Aβ burden has been reported to affect memory performance through its impact on sleep slow waves in elderly individuals (~75 years)[11]. In addition, the presence of preclinical Aβ plaque pathology, assessed through PET imaging or cerebrospinal fluid collection, is associated with fragmentation of the entire rest-activity cycle, i.e. encompassing both sleep and wakefulness[12]. Whether age-related alteration in brain structure may affect sleep–wake regulation of daytime brain activity is unknown, however.

Sleep and wakefulness are regulated by two fundamental processes: sleep homeostasis, which keeps track of time awake, and circadian rhythmicity, which temporally organizes physiology and behavior[13,14]. The strength of both processes seems to decrease with age, resulting in dampened dynamics of sleep–wake rhythms and reduced variations in brain activity both during sleep and prolonged wakefulness[13]. The generation of slow waves during sleep, which is associated with the dissipation of sleep need, is reduced in aging[15]. Likewise, cortical excitability[16], a basic aspect of brain function implicated in age-related cognitive decline[17,18], shows less variations during prolonged wakefulness. Age-related alterations in the regulation of sleep and wakefulness are not only associated with current cognition[19,20], but also predicts future cognitive trajectories, including the risk of developing dementia[12,21–23]. Importantly, some of the changes in sleep–wake regulation take place as early as in middle-aged individuals (>40 years)[15]. Whether the early alterations in sleep and wakefulness regulation and their potential cognitive consequences are systematically related to age-related alterations in brain structure remains unknown.

The goals of the present study were threefold. First, we assessed whether sleep–wake regulation of the awake and active brain is linked to age-related alterations in brain structure in healthy older individuals (50–70 years). We measured cortical excitability based on electroencephalographic (EEG) responses to transcranial magnetic stimulation (TMS) during a wake-extension protocol and hypothesized that the dynamics of cortical excitability during wakefulness would be related to both Aβ and tau burden, taking into account any potential neurodegeneration. We further investigated whether wake-dependent variations in cortical excitability are linked to cognitive fitness. Based on previous findings[16], we anticipated that cortical excitability dynamics would be associated with executive performance. Finally, we tested whether these putative links would be independent of Aβ and tau burden as well as neurodegeneration. We postulated that the inclusion of the three markers of brain structural integrity in our statistical models would at least decrease, if not remove, the association between cognition and wake-dependent cortical excitability dynamics.

Analyses reveal that frontal cortical excitability dynamics during prolonged wakefulness display a high variability across individuals, with some older people exhibiting preserved or young-like regulation profiles of cortical function. Preserved cortical excitability regulation profiles were associated with better cognitive performance, particularly in the executive domain which is central to successful cognitive aging. Importantly, the association between cognition and wake-dependent regulation of cortical function was independent of GM volume as well as Aβ and tau protein burden.

## Results

In a multi-modal cross-sectional study (Fig. 1a), 60 healthy and cognitively normal late middle-aged individuals (42 women; age range 50–69 years, mean ± SD = 59.6 ± 5.5 years; Table 1) underwent structural MRI to measure GM volume, as well as [18F] Flutemetamol and [18F]THK-5351 PET imaging to quantify Aβ and tau burden, respectively. Participants' cognitive performance while well-rested was assessed with an extensive neuropsychological task battery probing memory, attention, and executive functions. After a week of regular sleep–wake schedule, participants' habitual sleep was recorded in-lab under EEG to quantify slow waves generation during NREM sleep. A wake-extension protocol started on the following day and consisted of 20 h of continuous wakefulness to trigger a moderate realistic sleep–wake challenge. Wake-extension was conducted under strictly controlled constant routine conditions known to unmask the combined influences of sleep homeostasis and of the circadian system by removing or reducing potential biases from light exposure, physical activity, food intake or room temperature[14]. Cortical excitability over the frontal cortex was measured five times over the 20 h protocol, using TMS combined with an electroencephalogram (TMS-EEG) (Fig. 1b–d)[16,24]. Mean time between wake-extension protocol and cognitive assessment was 30.6 ± 37.7 days. Mean time between wake-extension protocol and brain structural integrity assessments was 56.7 ± 78.9 days for MRI, 121.3 ± 97.1 days for Aβ-PET, and 116.3 ± 108.3 days for Tau-PET.

Whole-brain GM volume was 41.04 ± 3.73% of total intracranial volume and showed the expected reduction with increasing age ($F_{1,56} = 5.86$, $p = 0.02$, semi-partial $R^2$ ($R^2_{\beta*}$) = 0.10; Supplementary Fig. 1a). Mean standardized uptake value ratio (SUVR) for Aβ and tau burden were respectively 1.16 ± 0.08 and 1.32 ± 0.10 and both showed a statistical trend for a positive association with age (Aβ: $F_{1,56} = 3.74$, $p = 0.06$; Tau: $F_{1,56} = 3.71$, $p = 0.06$; Supplementary Fig. 1b, c). Furthermore, whole-brain Aβ burden was strongly associated with whole-brain tau burden ($F_{1,55} = 16.00$, $p = 0.0002$, $R^2_{\beta*} = 0.23$; Supplementary Fig. 1d).

**Cortical excitability dynamics during wakefulness extension.** We first investigated wake-dependent changes in cortical excitability during the protocol, using a generalized linear mixed model (GLMM) including random intercept and repeated measurement autoregression [AR(1)]. Cortical excitability during prolonged wakefulness underwent significant changes with time awake after adjusting for age, sex, and education (GLMM, main effect of circadian phase; $F_{4,234.1} = 4.29$, $p = 0.0023$, $R^2_{\beta*} = 0.07$; Fig. 2a). Post hoc analyses revealed a global decrease from the

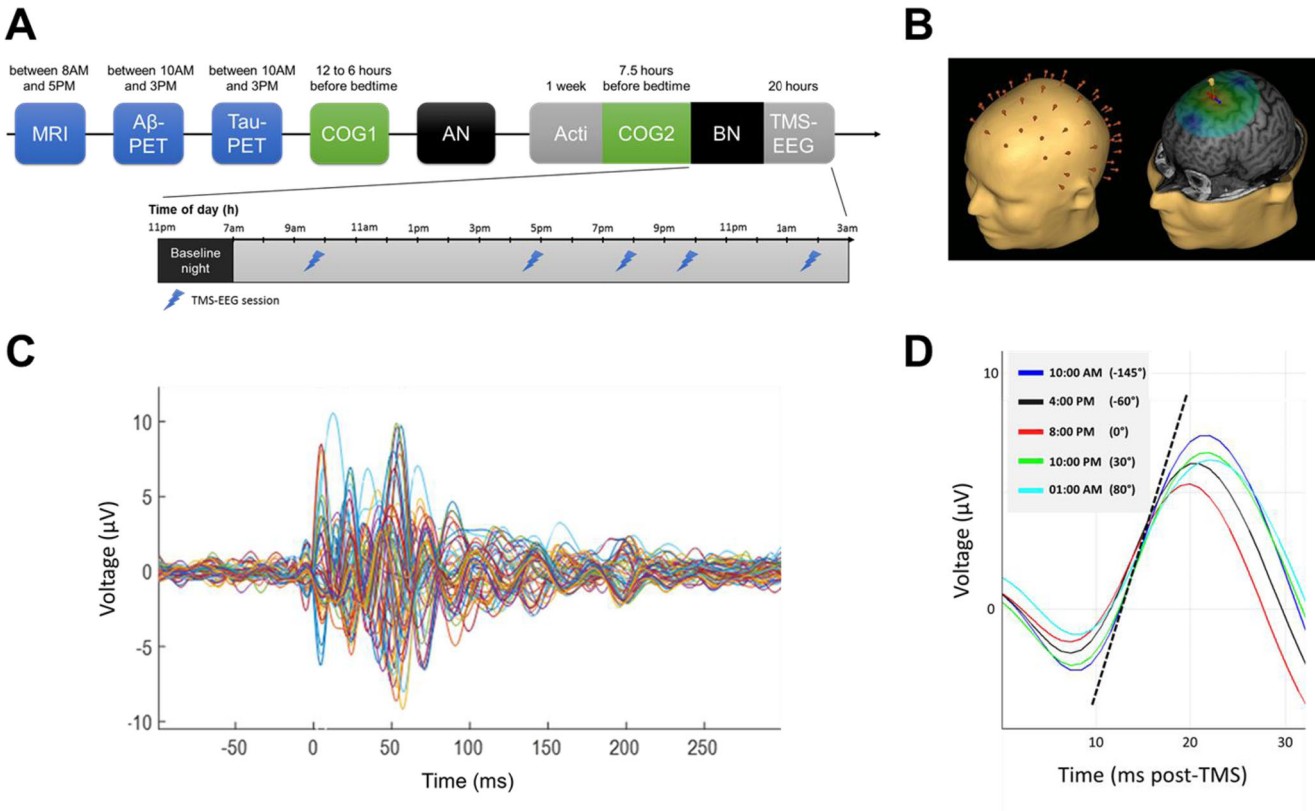

**Fig. 1** Study design and cortical excitability assessment. **a** Overview of the whole experimental protocol and the timing of the different steps for a representative subject with bedtime at 11:00PM and wake time at 07:00AM. AN: adaptation night with polysomnography to screen for sleep apnea; BN: baseline night under EEG recording. **b** Cortical excitability over the frontal cortex was assessed using neuronavigation-based TMS coupled to EEG. *Left*: reconstructed head with electrodes position; *Right*: representative location of TMS coil and stimulation hotspot with electric field orientation. **c** Butterfly plot of TMS-evoked EEG response over the 60 electrodes (−100 ms pre-TMS to 300 ms post-TMS; average of ~250 trials). **d** Representative TMS-evoked EEG potential (0–32 ms post-TMS) in the five TMS-EEG sessions with indicative clock time and circadian phase (15° = 1 h). Cortical excitability was computed as the slope (µV/ms) of the first component of the TMS-evoked EEG response at the electrode closest to the hotspot (dotted line: example for 10:00AM session).

beginning to the end of the protocol, with significant differences between the second and last TMS-EEG sessions ($p_{adj} = 0.007$), and between the third and last sessions ($p_{adj} = 0.02$). Visual inspection of individual data indicated, however, an important variability in cortical excitability values and in cortical excitability dynamics. The majority of subjects ($N = 35$, 25 women) displayed an overall decrease in cortical excitability throughout the protocol, whereas ~40% of the sample ($N = 25$, 17 women) exhibited an overall increase in cortical excitability, similar to what was previously reported in young adults[24,25].

To account for this variability, we summarized cortical excitability dynamics at the individual level using a single value consisting in the regression coefficient of a linear fit across the five TMS-EEG measurements (Fig. 2b). Individual residuals indicated that regression quality was good overall and regression coefficients reflected the differences between the first and last sessions of the protocol in most subjects (Supplementary methods). Therefore, the frontal cortical excitability profile (CEP) obtained through the regression fit across the protocol epitomizes cortical excitability dynamics and does not reflect random signal variations. We qualified individuals with positive regression coefficient as young-like CEP, and those with negative regression coefficient as old-like CEP. When considered as two separate groups, young-like and old-like CEP displayed distinct temporal patterns [GLMM, group × circadian phase interaction; $F_{4,234.1} = 13.69$, $p < 0.0001$, $R^2_{\beta*} = 0.19$; significant between-group post-hoc at circadian phase 30° ($p_{adj} = 0.02$) and 80° ($p = 0.01$);

Supplementary Fig. 2], but did not differ in any of the demographic or sleep–wake related variables reported in Table 1 ($p > 0.05$). Importantly, frontal CEP was considered as a continuous variable for all statistical analyses reported below and we did not consider young-like and old-like CEP as separate groups.

**CEP and sleep slow waves generation.** We then confronted the validity of frontal CEP as a measure of sleep–wake regulation by testing its association with slow waves generation during sleep[26]. We found that frontal CEP was significantly related to slow wave energy (SWE), a cumulative measure of slow waves generated during NREM sleep, both in the slower range (0.75–1 Hz, $F_{1,55} = 5.35$, $p = 0.02$, $R^2_{\beta*} = 0.09$, Fig. 3a) and in the higher range (1.25–4 Hz, $F_{1,55} = 5.47$, $p = 0.02$, $R^2_{\beta*} = 0.09$, Fig. 3b), such that young-like frontal CEP was associated with increased, and presumably preserved SWE. We further found that SWE in the lower 0.75–1 Hz range was expectedly associated with GM volume ($F_{1,53} = 7.90$, $p = 0.007$, $R^2_{\beta*} = 0.13$), as well as with whole-brain Aβ burden ($F_{1,53} = 5.15$, $p = 0.03$, $R^2_{\beta*} = 0.09$, Fig. 3c), confirming previous reports[11,27], but was not linked to whole-brain tau burden ($F_{1,53} = 1.26$, $p = 0.27$). In contrast, CEP was not associated with any of the brain structural integrity measures (GM: $F_{1,53} = 0.18$, $p = 0.67$; Aβ: $F_{1,53} = 0.16$, $p = 0.69$; Tau: $F_{1,53} = 0.48$, $p = 0.49$). Frontal CEP relates therefore to a gold standard measure of sleep homeostasis known to decline in aging[13,15] and to be

**Table 1 Sample characteristics (mean ± SD).**

|  | $n = 60$ |
| --- | --- |
| Sex | 42w/18 m |
| Age | 59.6 ± 5.5 |
| Education | 15.4 ± 3.2 |
| Right-handed | 59 |
| Ethnicity | Caucasian |
| Dementia rating scale | 142.1 ± 2.3 |
| Raven's progressive matrices | 50.5 ± 4.9 |
| Mill Hill vocabulary scale | 26.9 ± 3.9 |
| Body mass index (kg/m²) | 24.6 ± 2.9 |
| Anxiety | 2.6 ± 2.5 |
| Mood | 4.4 ± 4.7 |
| Caffeine (cups/day) | 3.6 ± 1.9 |
| Alcohol (doses/week) | 3.9 ± 4.0 |
| Treated for hypertension (stable > 6 months) | 7 |
| Treated for hypothyroidism (stable > 6 months) | 12 |
| Systolic blood pressure (mmHg) | 118.74 ± 11.62 |
| Sleep quality | 5.1 ± 3.0 |
| Daytime sleepiness | 6.2 ± 3.9 |
| Chronotype | 53.8 ± 8.3 |
| Clock time of dim-light melatonin onset (hh:min, PM) | 08:20 ± 00:59 |
| In-lab baseline sleep duration (min, EEG) | 388.0 ± 44.3 |
| In-lab baseline sleep efficiency, including N1 stage (%, EEG) | 82.6 ± 9.6 |
| Baseline sleep time (hh:min, PM) | 10:47 ± 00:36 |
| Baseline wake time (hh:min, AM) | 06:46 ± 00:43 |
| Gray matter volume (% of total volume) | 41.04 ± 3.73 |
| [18F]Flutemetamol (SUVR) | 1.16 ± 0.08 |
| [18F]THK-5351 (SUVR) | 1.32 ± 0.10 |

Anxiety was measured by the 21-item Beck Anxiety Inventory[53]; mood by the 21-item Beck Depression Inventory II[54]; caffeine and alcohol consumption by self-reported questionnaires; sleep quality by the Pittsburgh Sleep Quality Index[55]; daytime sleepiness by the Epworth Sleepiness Scale[56]; chronotype by the Horne-Östberg questionnaire (no participants were extreme chronotypes, i.e. scores <30 or >70[57]). Systolic blood pressure was measured in-bed after laying down for >15 min and 1 to 2 h prior to bedtime

associated with Aβ burden[11], but is not significantly linked to the hallmarks of AD neuropathology.

**CEP and cognition**. We tested whether frontal CEP and NREM SWE were related to global, memory, attentional, and executive cognitive performance in GLMMs including sex, age, and education as covariates. NREM SWE was not associated with any of the cognitive measures (Supplementary Table 1). In contrast, CEP was significantly and positively associated with global cognitive composite score, after accounting for the expected effects of age and education ($F_{1,55} = 6.76$, $p = 0.01$, $R^2_{\beta*} = 0.11$, Table 2, Fig. 4a). In other words, individuals displaying preserved frontal cortical excitability dynamics, i.e. young-like positive CEP, had better overall cognitive performance, compared to those characterized by old-like negative CEP. Strikingly, we further found a specific and strong positive association between frontal CEP and performance in the executive domain ($F_{1,55} = 8.47$, $p = 0.005$, $R^2_{\beta*} = 0.13$, Table 2, Fig. 4b). By contrast, neither memory ($F_{1,55} = 0.41$, $p = 0.52$) nor attention ($F_{1,55} = 2.44$, $p = 0.12$) were associated with CEP (Table 2, Fig. 4c, d), suggesting that the link between CEP and global cognition mainly arises from the executive domain. Replacing CEP by the difference between the first and last TMS-EGG sessions led to similar statistical outcomes, ensuring that our findings are not a spurious consequence of the linear regression fit approach and thus reflect the sleep–wake-dependent regulation of the build-up of sleep need on basic brain function (Supplementary methods; Supplementary Table 2).

**CEP and age-related changes in brain structure**. We found no association between the hallmarks of AD neuropathology and global, memory, attentional, and executive cognitive performance (Table 3). Critically, including region-specific brain structural integrity measures in the GLMMs seeking for associations between CEP and cognitive measures did not affect the statistical outcomes (Table 3); if anything, the link between CEP and executive performance became slightly stronger as suggested by semi-partial $R^2$ values ($R^2_{\beta*} = 0.13$ vs. $R^2_{\beta*} = 0.16$). Results were similar if we considered brain structural integrity measures only over the TMS stimulation target area, i.e. the superior frontal gyrus (Supplementary Table 3). The associations between CEP and cognition may therefore be independent of brain structural integrity measures.

## Discussion

Healthy aging is accompanied by a disruption of sleep and wakefulness regulation that participates to age-related cognitive decline. Sleep–wake modifications are rooted in part in age-related alterations in brain structure that can lead to AD. Previous investigations found that brain activity during sleep is related to Aβ burden, and that Aβ burden affects memory performance through modifications in brain activity during sleep[11]. This was however reported in elderly individuals (75.1 ± 3.5 years)[11], in which Aβ and tau proteins accumulation as well as neurodegeneration are likely relatively important, while daytime brain activity was not assessed. Here, we confirm that brain activity during sleep is associated with Aβ burden also in healthy late middle-aged individuals (59.6 ± 5.5 years). Yet, in this relatively large ($N = 60$) younger sample, slow waves generation during sleep is not associated with tau burden nor with cognitive measures including memory, attention, and executive functions. We demonstrate instead that, at around 60 years, wake-dependent variations of basic cortical function is associated with cognitive fitness, independently of Aβ and tau burden and GM volume.

Previous studies investigating the same stimulation area reported an overall increase in cortical excitability during prolonged wakefulness in young individuals (ref. [24], 40 h of sleep deprivation, six participants aged 25–41 years; ref. [25], 29 h of sleep deprivation, 24 participants aged 19–30 years), whereas only smaller variations were detected in healthy older individuals (ref. [16], 34 h of sleep deprivation, 13 young aged 19–27 years and 13 older participants aged 59–69 years). Here, in a larger sample size of older individuals, we showed that a mild wakefulness extension results in significant changes in cortical excitability with time awake. Over the entire sample, cortical excitability globally decreased from the beginning to the end of the protocol, with marked differences between the late-afternoon and the night time sessions. We further show that, amongst healthy older individuals, a large proportion (~40%) of participants displayed an overall wake-dependent increase in frontal cortical excitability which is similar to what was previously reported in young adults[24,25]. We interpret this as a sign of preserved, or young-like, temporal dynamics of cortical excitability, which corresponds to a positive CEP or linear regression coefficient over the entire protocol. Importantly, we find that positive CEP is correlated to increased slow waves generation during habitual sleep, suggesting that individuals with young-like CEP during wakefulness are characterized by a stronger and potentially preserved sleep homeostasis drive during sleep[13], compared to individuals with old-like CEP. This strongly suggests that frontal CEP is a measure of the active brain during wakefulness that reflects individuals' preservation of sleep–wake regulation processes.

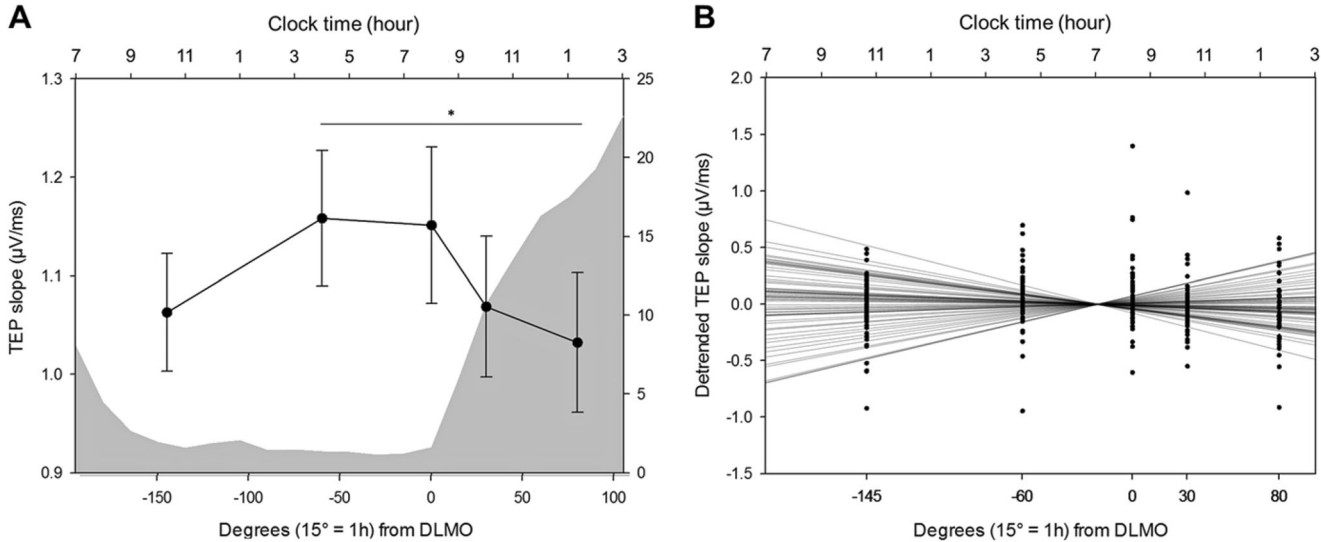

**Fig. 2 Cortical excitability dynamics as a marker of sleep–wake regulation processes. a** Average cortical excitability dynamics (mean ± SEM) during 20 h of prolonged wakefulness over the entire sample ($n = 60$). Gray background represents the average melatonin secretion profile (0° indicating dim-light melatonin onset, i.e. the beginning of the biological night; 15° = 1 h). *$p_{adj} < 0.01$. **b** Detrended cortical excitability values of all individuals and their respective linear regression lines across the five TMS-EEG measurements.

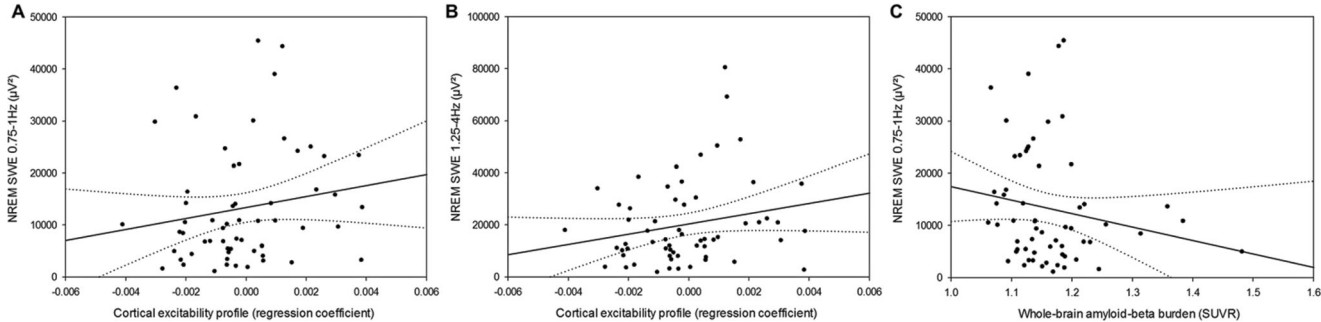

**Fig. 3 Cortical excitability, slow wave energy, and brain structural integrity. a** Positive association between CEP and cumulated frontal NREM SWE in the lower range (0.75–1 Hz) during habitual sleep ($n = 60$; $F_{1,55} = 5.35$, $p = 0.02$, $R^2_{\beta^*} = 0.09$). **b** Positive association between CEP and cumulated frontal NREM SWE in the higher range (1.25–4 Hz) during habitual sleep ($n = 60$; $F_{1,55} = 5.47$, $p = 0.02$, $R^2_{\beta^*} = 0.09$). **c** Negative association between NREM SWE (0.75–1 Hz range) and whole-brain amyloid-beta burden ($n = 60$; $F_{1,53} = 5.15$, $p = 0.03$, $R^2_{\beta^*} = 0.09$). Simple regressions were used only for a visual display and do not substitute the GLMM outputs. Dotted lines represent 95% confidence interval of these simple regressions.

Critically, we found that higher frontal CEP is associated with better overall cognitive performance, demonstrating a significant relationship between cortical excitability dynamics during wakefulness and cognition, which are both measured in an active and awake brain. In-depth cognitive phenotyping showed that this relationship was mainly driven by the performance in the executive domain. Executive functions refer to high-order cognitive processes (flexibility, inhibition, updating, etc.) needed for behavioral adjustment according to ongoing goals when facing new or complex situations[28]. Variations in executive performance assessed during sleep deprivation were previously found to be associated with cortical excitability dynamics in older and younger individuals[16]. Here, we further show that cognitive ability, as an individual trait measured outside a sleep deprivation protocol, is significantly associated with sleep–wake regulation of frontal cortical excitability. Executive functions influence other cognitive domains and are often seen as central in age-related cognitive decline to remain adapted to the environment and sustain day-to-day functioning in complete autonomy[29]. In addition, executive functions are considered to depend mainly on the frontal cortex processes, and their underlying cortical

networks undergo important changes in healthy aging[30]. These results therefore suggest that preserved frontal CEP may constitute a marker of cognitive fitness in aging, and particularly so in the executive domain.

Despite the relative youth of our sample (~60 years), the absence of association between cognitive measures and brain integrity markers of GM volume and protein burdens may appear surprising. Longitudinal studies have shown, however, that the association between some aspects of cognition and brain structure was especially apparent in participants aged 65 years and over[31]. Subtle differences in cognitive performance in relation to proteins accumulation or GM reduction may therefore not appear in our sample. This may also underlie the absence of significant association between sleep slow waves generation and tau burden that has been recently reported in an older sample of healthy individuals (73.8 ± 5.3 years)[10]. Furthermore, our sample is biased towards individuals with higher education (mean 15.5 ± 3.2 years) and with cognitive and brain reserves that may compensate for early alteration of brain structural integrity[3,32]. Alternatively, the composite scores for each cognitive domain may not be sensitive enough to be related to these early brain alterations. Nonetheless,

**Table 2 Associations between CEP and cognitive composite scores of global and domain-specific performance adjusted for age, sex, and education.**

|  | Global performance (Z-score) | Memory (Z-score) | Attentional (Z-score) | Executive (Z-score) |
|---|---|---|---|---|
| CEP | $F_{1,55} = 6.76$ $p = 0.01$ $R^2_{\beta^*} = 0.11$ | $F_{1,55} = 0.41$ $p = 0.52$ | $F_{1,55} = 2.44$ $p = 0.12$ | $F_{1,55} = 8.47$ $p = 0.005$ $R^2_{\beta^*} = 0.13$ |
| Age | $F_{1,55} = 6.97$ $p = 0.01$ $R^2_{\beta^*} = 0.11$ | $F_{1,55} = 3.76$ $p = 0.06$ | $F_{1,55} = 5.79$ $p = 0.02$ $R^2_{\beta^*} = 0.10$ | $F_{1,55} = 2.68$ $p = 0.11$ |
| Sex | $F_{1,55} = 0.01$ $p = 0.90$ | $F_{1,55} = 0.02$ $p = 0.90$ | $F_{1,55} = 0.05$ $p = 0.83$ | $F_{1,55} = 0.11$ $p = 0.74$ |
| Education | $F_{1,55} = 6.18$ $p = 0.02$ $R^2_{\beta^*} = 0.10$ | $F_{1,55} = 0.14$ $p = 0.71$ | $F_{1,55} = 3.60$ $p = 0.06$ | $F_{1,55} = 6.52$ $p = 0.01$ $R^2_{\beta^*} = 0.11$ |

Statistical outputs of generalized linear mixed models with cognitive scores as dependent measures, accounting for their respective data distribution profiles. $R^2_{\beta^*}$ corresponds to semi-partial $R^2$ in GLMMs

our findings indicate that sleep–wake regulation of brain activity during wakefulness, as measured by the dynamics of cortical excitability during a mild wakefulness extension, is either more sensitive than brain structural integrity markers to isolate associations with cognition in aging, or sensitive to aspects of cognition that undergo influences distinct from protein accumulations and neurodegeneration. Another explanation to our results might involve the soluble forms of Aβ and tau, as oligomers of both proteins were shown to alter neuronal function[33,34]. Currently these oligomers cannot be reliably measured in vivo and could not be accounted for in this experiment.

Furthermore, age-related molecular changes potentially underlying sleep need have been reported[35]. These may influence the local or global genetic and molecular machineries underlying circadian rhythmicity and sleep homeostasis, and in turn affect cortical excitability. Ageing modulates the impact of the basal forebrain, subcortical and brainstem ascending activating system on global brain activity[36]. In silico modeling of wake-dependent cortical excitability changes in young adults suggests that the fluctuations in the balance between excitation and inhibition within cortical networks may affect the observed variations during prolonged wakefulness[37]. However, this remains unexplored in healthy older individuals. It might also be the case that changes in cortical function stems from AD-related alteration of subcortical structures. In addition to the locus coeruleus[5], neurodegeneration of the suprachiasmatic nuclei, site of the master circadian clock, has been reported in AD[38] while network uncoupling in the suprachiasmatic nuclei is found in normal aging[39]. Future investigations should also examine whether cortical excitability dynamics, probed specifically over the frontal regions, are related to cognitive changes in aging when compared to cortical excitability dynamics probed over other parts of the brain and to other aspects of cognitive brain functions. Furthermore, the predictive value of frontal CEP assessment for subsequent cognitive decline and risk of developing dementia remains to be investigated in a longitudinal protocol.

This study presents several strengths. The use of TMS-EEG allows for a direct measure of cortical responsiveness while bypassing sensory systems and it mimics active brain processing without confounding biases. The prolonged wakefulness protocol is performed under strictly controlled constant routine conditions to control for multiple factors that could affect wakefulness and sleep parameters, such as light exposure or physical activity[14]. In addition, we performed a comprehensive multi-modal assessment of brain structure, including two PET scans and MRI for the hallmarks of AD pathophysiology, as well as an extensive neuropsychological investigation. Furthermore, sleep–wake history was controlled prior to wake extension and exclusion criteria ensured most risk factor favoring cognitive decline were not present in the sample (e.g. diabetes, smoking, alcohol abuse, depression, etc.)[40]. Finally, the relative young age of the participants reduces the accumulation of minor health issues associated with advanced age (e.g. diabetes, overweight, etc.), which can inherently affect findings in samples of elderly individuals. This research also has several limitations, however. Its cross-sectional nature does not allow us to comment on the future cognitive trajectory of participants. The reported effect sizes show that CEP does explain but a small part of variance, suggesting that its link with cognitive fitness is modest. This was expected given the numerous factors that affect cognitive trajectories[40] and the relatively young age and good overall health status of our sample. It is, in fact, quite remarkable that we were nonetheless able to isolate a link between wake-dependent cortical excitability dynamics and cognition in such sample, suggesting that it may be a very important link to successful cognitive aging. While the contribution of the homeostatic process is most obvious in our data, a longer protocol covering the whole circadian cycle would help disentangle the respective modulation of frontal CEP by the circadian system[14]. In addition, [18F]THK-5351 presents some unspecific binding, particularly around the fornix and basal ganglia[41]. We took this into account by excluding these portions of the brain from all tau burden measures. The observed links between [18F]THK-5351 uptake and both age and Aβ burden strongly support that our measure of tau burden, although potentially imperfect, was meaningful. Finally, we did not consider other age-related changes of brain integrity, such as cerebrovascular pathology, which are extremely common (up to 50%) as a mixed pathology in individuals with Alzheimer's dementia[42], and Lewy bodies pathology, which shares some genetic risk with AD[43].

Aging is the ultimate challenge that the brain has to face in order to maintain optimal cognition across the lifespan. The bidirectional detrimental interaction between disturbed sleep–wake regulation and AD pathogenesis suggest that sleep–wake interventions could be promising means to reduce the risk of dementia[6]. Here, we provide compelling evidence that sleep–wake regulation influences cognition in healthy older individuals, particularly in the executive domain, and beyond the changes in brain structural integrity that can ultimately lead to dementia. Since both sleep homeostasis and circadian rhythmicity show significant alteration as early as age 40[13,15], our results further reinforces the idea that sleep and wakefulness could be acted upon to improve individual cognitive health trajectory early in the lifespan. Our findings could have therefore implications for the understanding of brain mechanisms underlying the maintenance of cognitive health in normal and pathological aging, and for potential early intervention targets.

## Methods

**Study design and participants**. Between June 15, 2016, and July 28, 2018, healthy older individuals aged 50–70 years were enrolled for this multi-modal cross-sectional study after giving their written informed consent, and received a financial compensation. This research was approved by the Ethics Committee of the Faculty of Medicine at the University of Liège, Belgium. Exclusion criteria were: clinical symptoms of cognitive impairment (Dementia rating scale < 130; Mini mental state examination < 27); Body Mass Index ≤ 18 and ≥29; recent psychiatric history or severe brain trauma; addiction, chronic medication affecting the central nervous system; hypertension; smoking, excessive alcohol (>14 units/week) or caffeine (>5 cups/day) consumption; shift work in the past 6 months; transmeridian travel in the past 2 months; anxiety, as measured by the 21-item self-rated Beck Anxiety Inventory (score ≥ 10); depression, as assessed by the 21-item self-rated Beck Depression Inventory (score ≥ 14). Participants with stable treatment (for > 6 months) for hypertension and/or hypothyroidism were included in the study. Participants with

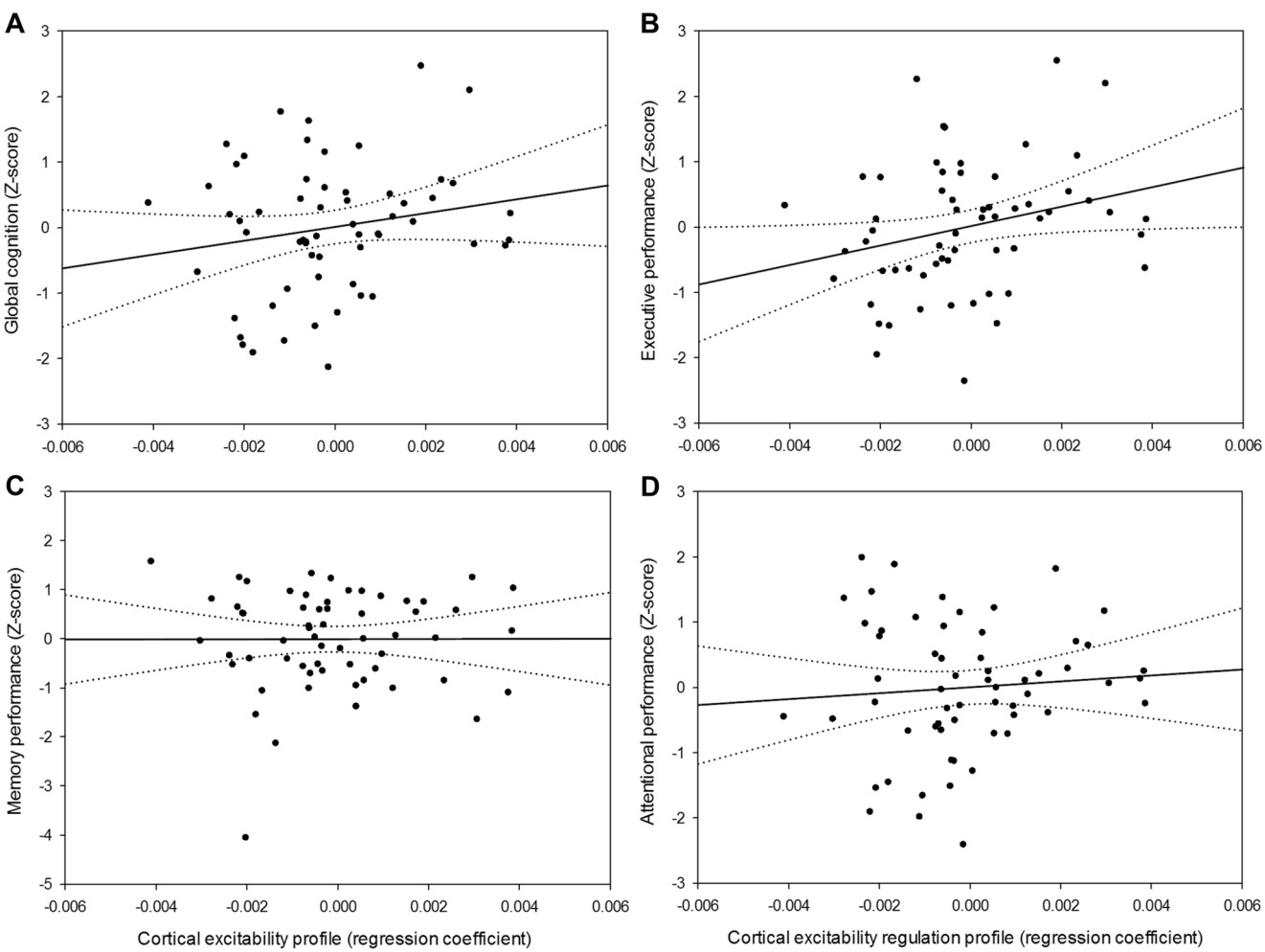

**Fig. 4 Relationships between CEP and cognition. a** Positive association between CEP and global cognition ($n = 60$; $F_{1,55} = 6.76$, $p = 0.01$, $R^2_{\beta*} = 0.11$). **b** Domain-specific positive association between CEP and performance to tasks probing executive functions ($n = 60$; $F_{1,55} = 8.47$, $p = 0.005$, $R^2_{\beta*} = 0.13$). **c** No significant association between CEP and memory performance ($n = 60$; $F_{1,55} = 0.39$, $p = 0.54$). **d** No significant association between CEP and attentional performance ($n = 60$; $F_{1,55} = 2.44$, $p = 0.12$). Simple regressions were used only for a visual display and do not substitute the GLMM outputs. Dotted lines represent 95% confidence interval of these simple regressions.

**Table 3 Associations between CEP and cognitive composite scores of global and domain-specific performance after accounting for global and region-specific brain structural integrity markers.**

|  | Global performance (Z-score) | Memory (Z-score) | Attentional (Z-score) | Executive (Z-score) |
|---|---|---|---|---|
| CEP | $F_{1,52} = 7.49$<br>$p = 0.009$<br>$R^2_{\beta*} = 0.13$ | $F_{1,52} = 0.39$<br>$p = 0.54$ | $F_{1,52} = 2.71$<br>$p = 0.11$ | $F_{1,52} = 9.71$<br>$p = 0.003$<br>$R^2_{\beta*} = 0.16$ |
| Age | $F_{1,52} = 7.48$<br>$p = 0.009$<br>$R^2_{\beta*} = 0.13$ | $F_{1,52} = 5.51$<br>$p = 0.02$<br>$R^2_{\beta*} = 0.10$ | $F_{1,52} = 5.22$<br>$p = 0.03$<br>$R^2_{\beta*} = 0.09$ | $F_{1,52} = 3.16$<br>$p = 0.08$ |
| Sex | $F_{1,52} = 0.04$<br>$p = 0.84$ | $F_{1,52} = 0.29$<br>$p = 0.59$ | $F_{1,52} = 0.01$<br>$p = 0.95$ | $F_{1,52} = 0.75$<br>$p = 0.39$ |
| Education | $F_{1,52} = 6.47$<br>$p = 0.01$<br>$R^2_{\beta*} = 0.11$ | $F_{1,52} = 0.99$<br>$p = 0.32$ | $F_{1,52} = 3.22$<br>$p = 0.08$ | $F_{1,52} = 5.82$<br>$p = 0.02$<br>$R^2_{\beta*} = 0.10$ |
| Region-specific GM volume | $F_{1,52} = 0.11$<br>$p = 0.75$ | $F_{1,52} = 3.54$<br>$p = 0.07$ | $F_{1,52} = 0.04$<br>$p = 0.85$ | $F_{1,52} = 0.10$<br>$p = 0.75$ |
| Region-specific Aβ burden | $F_{1,52} = 0.14$<br>$p = 0.71$ | $F_{1,52} = 1.96$<br>$p = 0.17$ | $F_{1,52} = 0.02$<br>$p = 0.88$ | $F_{1,52} = 0.02$<br>$p = 0.90$ |
| Region-specific Tau burden | $F_{1,52} = 1.17$<br>$p = 0.28$ | $F_{1,52} = 0.66$<br>$p = 0.42$ | $F_{1,52} = 1.07$<br>$p = 0.31$ | $F_{1,52} = 2.51$<br>$p = 0.12$ |

Statistical outputs of generalized linear mixed models with cognitive composite scores as dependent measures, accounting for their respective data distribution profiles. When considering global cognitive performance, region-specific Aβ and tau burden as well as GM density refer to whole-brain values. $R^2_{\beta*}$ corresponds to semi-partial $R^2$ in GLMMs.

sleep apnea (apnea-hypopnea index ≥ 15/h) were excluded based on an in-lab adaptation and screening night of polysomnography. One participant was excluded from the sample for all analyses because of outlier values on both PET assessments (>6 standard deviations from the mean). Demographic characteristics of the final study sample are described in Table 1.

**Magnetic resonance imaging.** High-resolution structural MRI was performed on a 3-T MR scanner (MAGNETOM Prisma, Siemens). For each participant, multi-parameter mapping volumes (i.e. T1-weighted, proton density-weighted, magnetization transfer (MT)-weighted) were acquired. We estimated individuals' total intracranial volume and whole-brain GM volume based on the MT-weighted image, using the SPM12 toolbox (https://www.fil.ion.ucl.ac.uk/spm/). For regional quantification of GM, volumes of interest were first determined using the Automated Anatomical Labeling atlas (AAL2)[44]. MT-weighted images were spatially normalized into a study-specific template with the Diffeomorphic Anatomical Registration Through Exponentiated Algebra (DARTEL) toolbox[45]. Volumes of interest were then applied on segmented normalized MT-weighted images and combined to extract GM volume in brain regions underlying each cognitive domain (Supplementary Table 4).

**PET imaging.** Aβ-PET imaging was performed with [18F]Flutemetamol, and tau-PET imaging was done with [18F]THK-5351. For both radiotracers, all participants received a single dose of their respective radioligands in an antecubital vein (target dose 185 MBq). Aβ-PET image acquisitions started 85 min after injection, and four frames of 5 min were obtained, followed by a 10-min transmission scan. For tau-PET, transmission scan was acquired first and dynamic image acquisitions started immediately after injection, consisting in 32 frames (with increasing time duration). All PET images were reconstructed using filtered back-projection algorithm including corrections for measured attenuation, dead time, random events, and scatter using standard software (ECAT 7.1, Siemens/CTI, Knoxville, TN). Motion correction was performed using automated realignment of frames without reslicing. A PET sum image was created using all frames for Aβ-PET, and using the four frames corresponding to the time window between 40 and 60 min for tau-PET. PET sum images were reoriented manually according to MT-weighted structural MRI volume and coregistered to structural MRI using the MT-weighted volume. PET sum images were further corrected for partial volume effect (PETPVC toolbox, iterative Yang method[46]) and spatially normalized using the MRI study-specific template. SUVR was calculated using the cerebellum GM as the reference region. The volumes of interest used for GM analysis were applied to normalized PET sum images to estimate regional SUVR of each radiotracer in cognitive domain-specific regions. [18F]THK-5351 radiotracer shows some unspecificity for tau, particularly over the basal ganglia, which was taken into account by excluding basal ganglia from all computations of PET SUVR values for tau-PET.

**Cognitive assessment.** Upon arrival for the wake-extension protocol and prior to being placed in dim-light (~7.5 h before habitual bedtime, corresponding to ~3:30PM for a representative subject with bedtime at 11:00PM), participants were administered the first part (~1 h) of the extensive neuropsychological assessment including: (1) Mnemonic Similarity Task; (2) Category Verbal Fluency (letter and animals); (3) Digit Symbol Substitution Test; (4) Visual N-Back (1−, 2−*, and 3-back variants); and (5) Choice Reaction Time. On another day while well-rested and during the day (from 12 to 6 h before habitual bedtime, i.e. between 11:00AM and 5:00PM for the same representative subject), the second part of the neuropsychological assessment was administered. This ~1.5 h session included: (1) Direct and Inverse Digit Span; (2) Free and Cued Selective Reminding Test; (3) Stroop Test; (4) Trail Making Test (part A and B); and (5) D2 Attention Test. The memory function composite score included Free and Cued Selective Reminding Test (sum of all free recalls) and Mnemonic Similarity Task (recognition memory score). The executive function composite score comprised verbal fluency tests (2-min score for letter and animal variants), the digit span (inverse order), Trail Making Test (part B), N-Back (3-back variant), and Stroop (number of errors for interfering items). The attentional function composite score included Digit Symbol Substitution Test (2-min score), Trail Making Test (part A), N-Back (1-back variant), D2 (Gz - F score), and Choice Reaction Time (reaction time to dissimilar items). We computed a composite score for each cognitive domain based on the sum of Z-scores on domain-related tasks, with higher scores reflecting better performance. Composite score for global cognitive performance score consisted of the standardized sum of the domain-specific composite scores.

**Sleep assessment and spectral power analysis.** As part of the screening process, participants first performed an in-lab adaptation and screening night to minimize the disrupting effect caused by sleeping in a novel environment[47], which might have otherwise affected the sleep parameters subsequently assessed during the baseline night. Then, for 7 days prior to the wake-extension protocol, participants followed a regular sleep–wake schedule (±30 min), in agreement with their preferred bed and wake-up times. Compliance was verified using sleep diaries and wrist actigraphy (Actiwatch©, Cambridge Neurotechnology, UK). Aside from the fixed sleep–wake schedule, participants were also instructed to abstain from intense physical exercise for the last 3 days of fixed-schedule circadian entrainment (i.e.

right before the wake-extension protocol; exercise/fitness levels were not controlled for more specifically). The day before the wake-extension protocol, participants arrived to the laboratory 8 h before their habitual bedtime and were kept in dim light (<5 lux) for 6.5 h preceding bedtime. Their habitual sleep was then recorded in complete darkness under EEG (baseline night, Fig. 1a). Baseline night data were acquired using N7000 amplifiers (EMBLA, Natus Medical Incorporated, Planegg, Germany). The electrode montage consisted of 11 EEG channels (F3, Fz, F4, C3, Cz, C4, P3, Pz, P4, O1, O2), two bipolar electrooculograms, and two bipolar electromyograms. Scoring of baseline night in 30-s epochs was performed automatically using a validated algorithm (ASEEGA, PHYSIP, Paris, France)[48]. An automatic artifact detection algorithm with adapting thresholds[49] was further applied on scored data. Power spectrum was computed for each channel using a Fourier transform on successive 4-s bins, overlapping by 2-s., resulting in a 0.25 Hz frequency resolution. The night was divided into 30 min periods, from sleep onset until lights on. For each 30 min period, SWE was computed as the sum of generated power in the delta band, both for the lower range (0.75–1 Hz) and higher range (1.25–4 Hz), during all the NREM 2 and NREM 3 epochs of the given period, after adjusting for the number of NREM 2 and 3 epochs to account for artefacted data. As the frontal regions are most sensitive to sleep–wake history[13], SWE was considered over the frontal electrodes (F3, Fz, F4).

**Wake-extension protocol.** The wake-extension protocol followed the baseline night and consisted of 20 h of continuous wakefulness under strictly controlled constant routine conditions, i.e. in-bed semi-recumbent position (except for scheduled bathroom visits), dim light <5 lux, temperature ~19 °C, regular isocaloric food intake, no time-of-day information, and sound-proofed rooms. The protocol schedule was adapted to individual sleep–wake time, and lasted up to the theoretical mid-sleep time (e.g. ca. 07:00 AM-03:00 AM) to create a moderate wakefulness extension challenge. Hourly saliva samples were collected for subsequent melatonin assays, allowing a posteriori data realignment and interpolation based on individual endogenous circadian timing. Importantly, participants were not informed about the schedule of the different events included in the wake-extension protocol (e.g. number and timing of scheduled TMS-EEG sessions, saliva samples collections, food intakes, etc.), nor about the exact duration of the protocol in order to avoid motivational/expectancy biases which might interfere with the wake-dependent effects on our measurements[50]. They were told during recruitment that the wake-extension protocol lasted for around 20 h, until the middle of the night. Cortical excitability over the frontal cortex was measured 5 times throughout the protocol, using TMS-EEG, with increased frequency around the circadian wake-maintenance zone, as it represents a critical period around which the interplay between sleep homeostasis and the circadian system show important changes. Participants were instructed that that the protocol included a few TMS-EEG session but were not informed about the exact number.

**TMS-EEG assessment.** One TMS-EEG session was performed prior to the wake-extension protocol to determine optimal stimulation parameters (i.e. location, orientation, and intensity) that allowed for EEG recordings free of muscular and magnetic artifacts. As in previous experiments[16,24,25], the target location was in the superior frontal gyrus. For all TMS-EEG recordings, pulses were generated by a Focal Bipulse 8-Coil (Nexstim, Helsinki, Finland). Interstimulus intervals were randomized between 1900 and 2200 ms. TMS-evoked responses were recorded with a 60-channel TMS-compatible EEG amplifier (Eximia, Helsinki, Finland), equipped with a proprietary sample-and-hold circuit which provides TMS artifact free data from 5 ms post stimulation. Electrooculogram was recorded with two additional bipolar electrodes. EEG signal was band-pass filtered between 0.1 and 500 Hz and sampled at 1450 Hz. Before each recording session, electrodes impedance was set below 5 kΩ. Each TMS-EEG session included ~250 trials (mean = 252 ± 15). Auditory EEG potentials evoked by the TMS clicks and bone conductance were minimized by diffusing a continuous white noise through earphones and applying a thin foam layer between the EEG cap and the TMS coil. A sham session, consisting in 30 TMS pulses delivered parallel to the scalp with noise masking, was administered to verify the absence of auditory EEG potentials. TMS-EEG data were preprocessed as previously described[16] in SPM12 implemented in MATLAB2013a (The Mathworks Inc., Natick, MA). In brief, TMS-EEG data underwent semi-automatic artifacts rejection, low-pass filtering at 80 Hz, down-sampling to 1000 Hz, high-pass filtering at 1 Hz, splitting into epochs spanning −101 and 300 ms around TMS pulses, baseline correcting (from −101 to −1 ms pre-TMS), and robust averaging. Cortical excitability was computed as the slope at the inflexion point of the first component of the TMS-evoked EEG potential on the electrode closest to the stimulation hotspot. For each participant, CEP value was defined as the regression coefficient of a linear fit (first order polyfit MATLAB function) across the 5 TMS-EEG measurements at the individual level.

**Melatonin assessment.** Salivary melatonin was measured by radioimmunoassay. The detection limit of the assay for melatonin was 0.8 ± 0.2 pg/l using 500 μl volumes. Dim-light melatonin onset times were computed for each participant using the hockey-stick method, with ascending level set to 2.3 pg/ml (Hockey-Stick software v1.5)[51]. The circadian phase of all TMS-EEG data points was estimated relative to individual dim-light melatonin onset time (i.e. phase 0°; 15° = 1 h).

Based on this, cortical excitability measures were resampled following linear interpolation at the theoretical phases of the TMS-EEG sessions of the protocol ($-145°$, $-60°$, $0°$, $30°$, $80°$).

**Statistics and reproducibility.** Statistical analyses were performed using GLMMs in SAS 9.4 (SAS Institute, Cary, NC). Dependent variables distribution was first determined using allfitdist function in MATLAB and GLMMs were adjusted accordingly. All statistical models were adjusted for age, sex and education. Statistical significance was set at $p < 0.05$. Simple regressions were used for visual display only and not as a substitute of the full GLMM statistics. Degrees of freedom were estimated using Kenward-Roger's correction. P-values in post-hoc contrasts (difference of least square means) were adjusted for multiple testing with Tukey's procedure. Semi-partial $R^2$ ($R^2_{\beta*}$) values were computed to estimate effect size of significant fixed effects in all GLMMs[52].

**Reporting summary.** Further information on research design is available in the Nature Research Reporting Summary linked to this article.

## Code availability

The authors declare that the codes used for data processing and/or statistical analyses are available from the corresponding author upon request.

## Data availability

The authors declare that the data supporting the findings of this study are available from the corresponding author upon request.

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

## Acknowledgements

The authors thank M. Blanpain, M. Cerasuolo, E. Lambot, C. Hagelstein, S. Laloux, E. Balteau, A. Claes, C. Degueldre, B. Herbillon, P. Hawotte, and B. Lauricella for their help in different steps of the study. M.V.E., P.G., C.S., C.P., C.B., F.C., G.V. are supported by the FNRS-Belgium. G.G. was supported by Wallonia Brussels International (WBI) and Fondation Léon Fredericq (FLF). The study was supported by Fonds National de la Recherche Scientifique (FRS-FNRS, F.4513.17, T.0242.19, and 3.4516.11, Belgium), Actions de Recherche Concertées (ARC SLEEPDEM 17/27-09) of the Fédération Wallonie-Bruxelles, University of Liège (ULiège), Fondation Simone et Pierre Clerdent, European Regional Development Fund (ERDF, Radiomed Project). [18F]Flutemetamol doses were provided and cost covered by GE Healthcare Ltd (Little Chalfont, UK) as part of an investigator sponsored study (ISS290) agreement. This agreement had no influence on the protocol and results of the study reported here.

## Author contributions

E.S., P.M., C.P., C.B., F.C., and G.V. designed the experiment. M.V.E., J.N., D.C., P.V.G., P.G., V.M., C.S., G.G., G.B., X.P., E.T., D.M., C.L.G., E.C., A.L., E.S., P.M., M.A.B., C.P., C.B., F.C., and G.V. helped in data acquisition, analysis, and interpretation. E.S., A.L., P.M., and C.P. provided administrative, technical, or material support. M.V.E,, D.C., and G.V. wrote the manuscript. M.V.E., J.N., D.C., P.V.G., P.G., V.M., C.S., G.G., G.B., X.P., E.T., D.M., C.L.G., E.C., A.L., E.S., P.M., M.A.B., C.P., C.B., F.C., and G.V. contributed to manuscript revising.

## Competing interests

The authors declare no competing interests.
