## [Peer Review File · Communications Biology]

Reviewers' comments:

Reviewer #1 (Remarks to the Author):

The authors have conducted an elegant study that aims to test how brain function during wakefulness is linked to brain structure in younger vs older individuals, including by assessing tau and amyloid burden using PET imaging. The finding of preserved cortical excitability being linked to cognitive function is highly interesting. The manuscript is furthermore well-written, the statistical analyses are appropriate but there are some issues that need to be addressed. There are for instance some details in the methods section that need to be further elaborated on, and which would strengthen the validity of the data, especially for follow-up studies and attempts at validation by other research groups.

Comment: The abstract as well as the last part of the introduction are a bit vague. Specifically, it should be further clarified what the authors mean by brain integrity. Furthermore, also later in the text, the parameter "sleep-wake regulation of brain activity during wakefulness", which the authors measure, should be much more clearly defined.

Comment: Was an adaption night used during which participants were adapted to the laboratory/testing environment? Otherwise it may be hard to rule out differences due to the "night watch" effect (see Tamaki et al., Curr Biol 2016, PMID: 27112296). This should otherwise be added as a caveat, given that sleep onset latency and other sleep parameters also are altered by age, and thus an increase in these negative sleep effects could interact with the sleep-disrupting experience of sleeping in a completely novel environment.

Comment: Prior to study participation, participants were presumably informed about the timing of the different events? If so, it is possible that the constant routine was not completely masked/devoid of time cues. Such details are often not highlighted in the methods section, even of the most cited or well-controlled constant routine studies.

Comment: Was ApoE genotype information available? This may clarify why the authors observe strong inter-individual differences. For instance, rs429358 and rs7412, or alternatively rs4420638 would be of interest to classify subjects into low vs high risk subjects. Similarly, was information available about their hereditary predisposition to any dementia or specifically AD?

Comment: For figure 1, please include a timeline for the study segments/events that occurred prior to the in-lab session.

Comment: The authors mention that the participants were of "relative young age", but they are in fact in advanced/late middle age. Thus, it is highly likely that over 50% of the participants had hypertension (c.f. <https://www.cdc.gov/nchs/products/databriefs/db289.htm>). There is however no information about the prevalence of hypertension diagnosis, current blood pressure or use of anti-hypertensive medication. The authors should include information about all these parameters.

Comment: Did the authors focus specifically on frontal CEP? Perhaps the authors could consider that CEP could be abbreviated F-CEP, to further indicate this?

Comment: Have the authors assessed whether the subjects with reduced vs increased cortical excitability over the course of the wakefulness protocol differ in their habitual sleep duration, wake-up time or chronotype?

Comment: When comparing with previous studies in the discussion, please clarify whether they also focused on specific areas for assessing cortical excitability during prolonged wakefulness, and whether their protocols were of similar wake duration.

Comment: Why were the cognitive tests done partly during the evening prior to the wake protocol, and partly during daytime on another separate day? What determined which test that should be done when? Did the authors take chronotype into account for the tests done late (or early) during the day? Late chronotypes (i.e. nightowls) are known to perform tests better in the evening, and vice versa for morning types. It would be helpful to include chronotype in the tests comparing cognitive performance against sleep parameters and cortical excitability.

Comment: During the wake-extension protocol, given that it was not long enough to assess circadian parameters, it would be good if the authors explained in the text why a dim-light with no physical activity condition was chosen. Also, was physical activity controlled for prior to participating in the in-lab study, i.e. was level of activity restricted somehow?

Comment: Have the authors double checked the value for grey matter shown in Table 1? It reads " 0.41 ± 0.04 " and therefore it seems like it's not been multiplied by a factor 100.

Comment: Given that Amyloid beta binding in brain by PET may also be affected by sleep (see Shokri-Kojori et al, Proc Natl Acad Sci U S A. 2018, PMID: 29632177), did the authors take time of day and prior sleep duration into account when the PET for amyloid beta and tau were done?

Reviewer #2 (Remarks to the Author):

The manuscript by Van Egroo et al., investigates the inter-relation of cortical excitability on cognitive fitness in elderly people. Goal of this study was to discern the role of cortical excitability and wake-dependent cycles from the role of structural neuroanatomical factors. The authors conducted a study combining different methods. Generally, the topic of the submitted work is very interesting and timely and the authors used many methods to try to corroborate their results and findings. I am not an expert in PET and sleep EEG, but I do see several severe problems in the interpretation of the study findings and the other methods being used. Below, I only detail my most severe concerns. In my opinion, these cannot be dealt with in a standard revision. Unfortunately, I have to recommend rejection of this manuscript.

1. One of the major findings of the study is that TMS-evoked EEG changes are specifically correlated with executive functions in elderly subjects. However, this is a very trivial finding considering that the authors applied TMS pulses above the superior frontal gyrus (SFG). The SFG is well-known to play an important role in the sort of cognitive control functions the authors investigated in their neuropsychological assessment used for the executive function score. The authors would have been in a much stronger position if other TMS stimulation sites would have been used as well.
2. Related to the above point the authors state that structural neuroanatomical factors did not provide such strong correlations. The authors took this as evidence that especially the neurophysiological excitability and not structural neuroanatomical factors are important to consider. However, looking at their „ROI“ analysis of the MRI data, the authors created a ROI that covers many more „executive functions areas“ than the areas they stimulated in the TMS session. The SFG was only one of 10 areas forming this ROI. This means that the neuroanatomical predictor uses information for which at best 10% is due to the neuroanatomical regions being target using the TMS protocol. Such a regressor MUST yield null effects (especially in comparison to the TMS results). In some way, the authors are

comparing apples with oranges when they claim that functional neuroanatomical aspects are much less predictive than neurophysiological processes.

3. I also have problems with the method used to examine cortical excitability. The TMS-EEG method used by the authors has, just recently, heavily been criticized (cf. Conde et al., 2019, *Neuroimage*; Siebner et al., 2019, *Brain Stimul*). It has been shown that the TMS-EEG protocol used is by no means able to really pinpoint cortical excitability. Therefore, the major outcome measure of this study is severely flawed.

Author's Response to Reviewers (and Editors):

Reviewer #1:

We thank Reviewer #1 for acknowledging the interest of the topic and for their relevant comments and suggestions, which were mostly directed towards methodological details and which improved the clarity of our text.

Comment #1: *'The abstract as well as the last part of the introduction are a bit vague. Specifically, it should be further clarified what the authors mean by brain integrity. Furthermore, also later in the text, the parameter "sleep-wake regulation of brain activity during wakefulness", which the authors measure, should be much more clearly defined.'*

Response #1: We revised the abstract to account for this comment. It now mentions our MRI-based index of grey matter volume as a brain structural integrity marker (Page 2, lines 8-9), and the cognitive functions assessed in our study (Page 2, lines 9-10). Furthermore, we modified throughout the text (including the abstract) the terms 'brain integrity' to clarify that we refer to modifications in brain structure as well as the terms 'sleep-wake regulation of brain activity during wakefulness' which was in most cases modified to 'wake-dependent regulation of cortical function'.

Comment #2: *'Was an adaption night used during which participants were adapted to the laboratory/testing enviroment? Otherwise it may be hard to rule out differences due to the "night watch" effect (see Tamaki et al., Curr Biol 2016, PMID: 27112296). This should otherwise be added as a caveat, given that sleep onset latency and other sleep parameters also are altered by age, and thus an increase in these negative sleep effects could interact with the sleep-disrupting experience of sleeping in a completely novel environment.'*

Response #2: As stated in the legend of Figure 1, an adaptation night (which we referred to as 'habituation night' in the previous version) was indeed performed before the baseline night in order to minimize the effects of sleeping in a novel environment on the sleep parameters subsequently assessed during the baseline night. This adaptation night was performed in the exact same environment as the baseline night and was also implemented to screen for our exclusion criteria related to sleep apnea using polysomnography. We modified Figure 1

accordingly in order to call it ‘adaptation night’, and we changed the legend to describe it as ‘adaptation night with polysomnography to screen for sleep apnea’. We also added the following lines in the methods:

Page 20, lines 4-7: ‘As part of the screening process, participants first performed an in-lab adaptation and screening night to minimize the disrupting effect caused by sleeping in a novel environment [47], which might have otherwise affected the sleep parameters subsequently assessed during the baseline night. Then, for 7 days prior to the wake-extension protocol, [...]’.

Comment #3: *‘Prior to study participation, participants were presumably informed about the timing of the different events? If so, it is possible that the constant routine was not completely masked/devoid of time cues. Such details are often not highlighted in the methods section, even of the most cited or well-controlled constant routine studies.’*

Response #3: As motivational or expectancy biases might interfere with the wake-dependent effects on our measurements, participants were not informed about the schedule of the events during the wake-extension protocol, nor about its exact duration (they were told during recruitment that the wake-extension protocol lasted for *around* 20 hours, until the middle of the night). Furthermore, all experimenters were asked to avoid giving any time-of-day information or any cues about the number of remaining scheduled events (i.e. number of remaining TMS-EEG sessions, saliva samples, etc.) throughout the entire data acquisition. We added those methodological details and modified the wake-extension protocol description accordingly in order to highlight these aspects:

Page 21, lines 11-16: ‘Importantly, participants were not informed about the schedule of the different events included in the wake-extension protocol, (e.g. number and timing of scheduled TMS-EEG sessions, saliva samples collections, food intakes, etc.), nor about the exact duration of the protocol in order to avoid motivational/expectancy biases which might interfere with the wake-dependent effects on our measurements [].’

[] Hull et al., 2003.

Comment #4: *‘Was ApoE genotype information available? This may clarify why the authors observe strong inter-individual differences. For instance, rs429358 and rs7412, or*

alternatively rs4420638 would be of interest to classify subjects into low vs high risk subjects. Similarly, was information available about their hereditary predisposition to any dementia or specifically AD?’

Response #4: ApoE genotype information is not available for this manuscript, even though having such information would probably improve explained variance and open new interesting analyses. We do not feel however that ApoE genotype is critical to the observed relationship between cognition and wake-dependent cortical excitability dynamics, as ApoE polymorphisms are mostly associated with the risk of amyloid-beta accumulation (which was not related to cortical excitability profile) and the risk of having sleep-disordered breathing (which was part of the exclusion criteria assessed during the screening night of polysomnography). Regarding the hereditary predisposition, we do not have other genetic information and we did not systematically ask participants for potential family history of dementia. The age range of our participants most probably rules out the familial form of autosomal dominant early-onset AD.

Comment #5: *‘For figure 1, please include a timeline for the study segments/events that occurred prior to the in-lab session.’*

Response #5: We modified figure 1a and figure 1 legend accordingly.

Comment #6: *‘The authors mention that the participants were of “relative young age”, but they are in fact in advanced/late middle age. Thus, it is highly likely that over 50% of the participants had hypertension (c.f. <https://www.cdc.gov/nchs/products/databriefs/db289.htm>). There is however no information about the prevalence of hypertension diagnosis, current blood pressure or use of anti-hypertensive medication. The authors should include information about all these parameters.’*

Response #6: We used the terms ‘of relative young age’ in order to contrast with the population studied in previous research cited in the text (> 65 years), and to highlight the fact that this particular age range (‘late middle age’) is an important aspect of our sample. We modified the text accordingly to reduce confusion.

Non-treated hypertension was an exclusion criteria, and we only included participants whose hypertension was treated with non-beta-blocker medications (with stable treatment for at least 6 months). Table 1 now includes the number of participants with stable treatment for hypertension, as well as hypothyroidism, and we added information to the method section:

Page 17, lines 13-14: ‘Participants with stable treatment (for > 6 months) for hypertension and/or hypothyroidism were included in the study.’

We also slightly modified one sentence in the discussion to account for this comment:

Page 15, lines 4-5: ‘Finally, the relative young age of the participants reduces the accumulation of minor health issues associated with advanced age (e.g. ~~hypertension~~, diabetes, overweight, etc.), which can inherently affect findings in samples of elderly individuals’.

Comment #7: *‘Did the authors focus specifically on frontal CEP? Perhaps the authors could consider that CEP could be abbreviated F-CEP, to further indicate this?’*

Comment #7: As mentioned several times in the text, we focused on frontal CEP because this region shows the most pronounced wake-dependent modulation of EEG measurements. We modified several sentences throughout the manuscript to add ‘frontal’ before ‘CEP’ in order to further indicate that our results pertain to the frontal region, and we rephrased part of the discussion to further elaborate on this issue:

Page 14, line 11: ‘Future investigations should also examine whether cortical excitability dynamics, **probed specifically over the frontal regions**, are related to cognitive changes in aging when compared to **cortical excitability dynamics probed over** other parts of the brain and to other aspects of cognitive brain functions. Furthermore, the predictive value of **frontal** CEP assessment for subsequent cognitive decline and risk of developing dementia remains to be investigated in a longitudinal protocol.’

Comment #8: *‘Have the authors assessed whether the subjects with reduced vs increased cortical excitability over the course of the wakefulness protocol differ in their habitual sleep duration, wake-up time or chronotype?’*

Response #8: As stated in the results section (Page 8, lines 2-3), increasing/decreasing frontal CEP was not associated with differences in any of the variables reported in Table 1, including

variables of habitual sleep duration, wake-up time, and chronotype. We rephrased to make it clearer:

Page 7-8, lines 22-3: ‘When considered as two separate groups, young-like and old-like CEP [...], but did not differ in any of the demographic **or sleep-wake related variables** reported in Table 1 ($p > 0.05$).’

Comment #9: *‘When comparing with previous studies in the discussion, please clarify whether they also focused on specific areas for assessing cortical excitability during prolonged wakefulness, and whether their protocols were of similar wake duration.’*

Response #9: All three previous studies probed the same stimulation area (i.e. superior frontal gyrus) for TMS-EEG, but differed in terms of population and/or wake duration. We modified accordingly to clarify these differences:

Page 11, lines 15-19: ‘Previous studies investigating the same stimulation area reported an overall increase in cortical excitability during prolonged wakefulness in young individuals ([24], 40h of sleep deprivation, 6 participants aged 25-41 years; [25], 29h of sleep deprivation, 24 participants aged 19-30 years), whereas only smaller variations were detected in healthy older individuals ([16], 34h of sleep deprivation, 13 young aged 19-27 years and 13 older participants aged 59-69 years).’

[24] Huber et al., 2013; [25] Ly et al., 2016; [16]: Gaggioni et al., 2019.

Comment #10: *‘Why were the cognitive tests done partly during the evening prior to the wake protocol, and partly during daytime on another separate day? What determined which test that should be done when? Did the authors take chronotype into account for the tests done late (or early) during the day? Late chronotypes (i.e. nightowls) are known to perform tests better in the evening, and vice versa for morning types. It would be helpful to include chronotype in the tests comparing cognitive performance against sleep parameters and cortical excitability.’*

Response #10: As described in the methods, cognitive tests were performed in two parts: the first part was done ~7.5 hours before habitual bedtime, on the day before the baseline and the wake-extension protocol, i.e. in the afternoon. For a representative subject with habitual bedtime at 11:00PM, this corresponds to ~3:30PM. Most importantly, as for all the data

acquired on the baseline day, baseline night, and wake-extension day, the timing was adjusted on the basis of participants' preferred sleep-wake schedule (strictly followed for 7 days), which ensured that every participants were cognitively tested at the same phase of their biological sleep-wake cycle. Given the extensive aspect of our neuropsychological task battery, the second part was done on a separate day. Again, in order to standardize our acquisitions, the second cognitive test session was always performed between 12 to 6 hours before bedtime (i.e. from ~11:00AM and ~5:00PM for the same representative subject). It is also important to note that, within the two cognitive sessions, the order of the tasks remained constant across all individuals to minimize inter-individual time-on-task effects. Finally, the allocation of each task to one or the other cognitive sessions was set to counterbalance not only the length of the session, but also the distribution of the cognitive load associated with each cognitive function (e.g. avoiding to have all memory-based tasks in the same session).

Chronotype was not an inclusion/exclusion criteria and was not considered to set the timing of the different parts of the protocol. The timing of the both cognitive test batteries, i.e. 7.5h and 6 to 12h before bedtime respectively, is not a period of important chronotype-dependent changes in cognitive performance, particularly given that none of our participants were extreme chronotypes (scores < 30 or > 70).

We modified the cognitive assessment description to include an example of the timing for a representative subject:

Page 19, lines 6-13: 'Upon arrival for the wake-extension protocol and prior to being placed in dim-light (~7.5 hours before habitual bedtime, corresponding to ~3:30PM for a representative subject with bedtime at 11:00PM), participants were administered the first part (~1h) of the extensive neuropsychological assessment including [...]. On another day while well-rested and during the day (from 12 to 6 hours before habitual bedtime, i.e. between 11:00AM and 5:00PM for the same representative subject), the second part of the neuropsychological assessment was administered.'

Comment #11: *'During the wake-extension protocol, given that it was not long enough to assess circadian parameters, it would be good if the authors explained in the text why a dim-light with no physical activity condition was chosen. Also, was physical activity controlled for prior to participating in the in-lab study, i.e. was level of activity restricted somehow?'*

Response #11: Light exposure and physical activity (but also food intake and environmental temperature) affect alertness and mask the impact of sleep homeostasis and of the circadian system. Given that our main aim was to assess the combined impact of these endogenous phenomenon on brain function in association with AD PET markers, we had to conduct the study in conditions freed from masking effects factor (i.e. constant routine). In addition, it is well-established that melatonin secretion is affected by both light exposition and physical activity. Since we used dim-light melatonin onset (DLMO) as a circadian phase marker (DLMO = phase 0°) for TMS-EEG data inter-individual alignment and interpolation, we had to assess melatonin secretion in conditions that are not biased by these factors. We included this aspect in the manuscript:

Page 11, lines 11-14: ‘Wake-extension was conducted under strictly controlled constant routine conditions known to ‘unmask’ the combined influences of sleep homeostasis and of the circadian system by removing or reducing potential biases from light exposure, physical activity, food intake or room temperature [14]’

In addition to the imposed sleep-wake schedule, physical activity was partly controlled for prior to participating to the wake-extension protocol. Indeed, for the last 3 days of fixed-schedule circadian entrainment (i.e. right before the in-lab protocol), participants were instructed to abstain from unusual physical exercise:

Page 20, lines 10-12: ‘Aside from the fixed sleep-wake schedule, participants were also instructed to abstain from unusual physical exercise for the last 3 days of fixed-schedule circadian entrainment (i.e. right before the in-lab wake-extension protocol).’

Comment #12: *‘Have the authors double checked the value for grey matter shown in Table 1? It reads “0.41 ± 0.04” and therefore it seems like it’s not been multiplied by a factor 100.’*

Response #12: It needed indeed to be multiplied by a factor 100. We modified accordingly and the grey matter values now correctly correspond to percentage of total intracranial volume.

Comment #13: *‘Given that Amyloid beta binding in brain by PET may also be affected by sleep (see Shokri-Kojori et al, Proc Natl Acad Sci U S A. 2018, PMID: 29632177), did the*

authors take time of day and prior sleep duration into account when the PET for amyloid beta and tau were done?’

Response #13: We did not strictly control for time of day or sleep duration prior to PET acquisitions in our study. However, for practical and administrative reasons, all PET sessions were done between 10:00AM and 3:00PM. Since PET scans were performed during week days in middle-aged individuals, it is unlikely that any of them had been engaged in social activity totally preventing them from getting any sleep prior to PET scans (none were shift workers). In addition, no participant reported a night without sleep before PET sessions or seemed abnormally sleepy. Total sleep deprivation has been reported to affect PET A β binding in 20 individuals aged 22 to 72 y, but we are not aware of studies reporting a link between A β or tau PET binding and objectively investigated sleep duration on the preceding night. Time-of-day has been reported to affect the dynamics of A β and tau content in the CSF, but we are not aware of any study reporting an impact of time-of-day on A β or tau burden investigated with PET.

Therefore, while we cannot formally rule out any time-of-day or prior sleep duration effect on the PET measures, given the conditions under which PET scans were performed, we consider unlikely that they significantly affected our PET measurements.

Reviewer #2:

We thank Reviewer #2 for highlighting the timeliness of our research topic and for mentioning the recent questions about the TMS-EEG methods. It gave us the opportunity to improve the clarity of our text.

However, we respectfully but most strongly disagree with Reviewer #2 when they state that our study is not suitable for publication on the basis of three main concerns. In the following lines, we show that, although relevant, the main issues raised by Reviewer #2 do not apply to our study. We especially stress that issue #3, which may have been the main driver for the reviewer's recommendation to reject our paper, does not apply to our research and does not compromise at all our findings.

Comment #1: *'One of the major findings of the study is that TMS-evoked EEG changes are specifically correlated with executive functions in elderly subjects. However, this is a very trivial finding considering that the authors applied TMS pulses above the superior frontal gyrus (SFG). The SFG is well-known to play an important role in the sort of cognitive control functions the authors investigated in their neuropsychological assessment used for the executive function score. The authors would have been in a much stronger position if other TMS stimulation sites would have been used as well.'*

Response #1: This first comment seems to result from a misunderstanding of the main goal and findings of our study. Our neurophysiological measure of interest resides in the temporal variations of cortical excitability values over the wake-extension protocol, which inherently involve homeostatic/circadian processes. We demonstrate that frontal cortical excitability dynamics is linked to executive abilities, not that frontal cortical excitability at any given time is related to executive performance. In other words, our goal goes beyond the mere functional role of the superior frontal gyrus that would be assessed with a single TMS-EEG session.

Furthermore, to the best of our knowledge, our findings constitute the first evidence of a link between wake-dependent variations in cortical excitability and cognition as a trait (i.e. not assessed when facing a sleep deprivation challenge) in humans. While other research teams mostly emphasized the impact of sleep parameters (such as sleep slow wave activity or sleep fragmentation) on cognition in older individuals, we demonstrate that the wake-dependent variation in the active brain, reflected in changes in cortical excitability, may constitute a new

marker of cognitive aging. The fact that we were able to observe such relationships in a cohort of healthy late middle-aged individuals supports a role of sleep-wake-dependent neuronal dynamics in the earliest stages of cognitive aging. We do consider that this finding is highly valuable and constitute a significant step forward for the growing body of literature pointing up a close association between cognitive trajectories across the lifespan and sleep-wake regulation parameters.

While we agree that probing another stimulation site would be of high interest to determine whether regional difference may exist in the link between cortical excitability dynamics and different aspects of cognition, we consider that this can only constitute a subsequent step built on the observations reported in the present study.

Comment #2: *'Related to the above point the authors state that structural neuroanatomical factors did not provide such strong correlations. The authors took this as evidence that especially the neurophysiological excitability and not structural neuroanatomical factors are important to consider. However, looking at their „ROI“ analysis of the MRI data, the authors created a ROI that covers many more „executive functions areas“ that the areas they stimulated in the TMS session. The SFG was only one of 10 areas forming this ROI. This means that the neuroanatomical predictor uses information for which at best 10% is due to the neuroanatomical regions being target using the TMS protocol. Such a regressor MUST yield null effects (especially in comparison to the TMS results). In some way, the authors are comparing apples with oranges when they claim that functional neuroanatomical aspects are much less predictive than neurophysiological processes.'*

Response #2: Again, the goal of our analysis may have been unclear to the reviewer. Our aim was to link frontal cortical excitability dynamics (as probed over the SFG) to performance over different cognitive domains, and not to performance to a task that would specifically recruit the SFG. For each cognitive function (i.e. memory, attention, and executive), we therefore used large networks coming from meta-analyses of the literature in order to account for the cognitive heterogeneity that defines the tasks used to build the cognitive composite scores.

In line with Reviewer #2's suggestion, we carried out supplemental analyses only including brain integrity marker in the SFG. These also yielded a significant association between CEP and executive performance ($F_{1,52} = 8.99$, $p = 0.0042$), but revealed no significant relationship between executive performance and SFG structural integrity markers ($A\beta$, tau, grey matter

volume), both unilaterally to include only the stimulation site ($A\beta$: $F_{1,52} = 0.34$, $p = 0.56$; tau: $F_{1,52} = 0.96$, $p = 0.33$; grey matter: $F_{1,52} = 0.09$, $p = 0.76$), and bilaterally ($A\beta$: $F_{1,52} = 0.31$, $p = 0.58$; tau: $F_{1,52} = 1.35$, $p = 0.25$; grey matter: $F_{1,52} = 0.04$, $p = 0.85$). Therefore, even when considering a much more specific ROI around our stimulation area, we do not find associations between structural aspects and cognition. These additional analyses are now included as supplementary materials (Supplemental table 4).

As stated in the discussion, we interpret our results in the context of our study sample which, unlike most studies in cognitive aging, comprises late middle-aged individuals thoroughly screened regarding their physical health, life habits, sleep-wake history, and cognitive status. In that context, we argue that being able to identify a link between cognition and wake-dependent modulation of cortical excitability provide significant evidence for the importance of the temporal organization of brain dynamics during wakefulness in early cognitive aging, beyond (and speculatively before) the structural aspects of these brain networks. This statement is further reinforced by the analyses suggested by Reviewer #2.

Comment #3: *'I also have problems with the method used to examine cortical excitability. The TMS-EEG method used by the authors has, just recently, heavily been criticized (cf. Conde et al., 2019, Neuroimage; Siebner et al., 2019, Brain Stimul). It has been shown that the TMS-EEG protocol used is by no means able to really pinpoint cortical excitability. Therefore, the major outcome measure of this study is severely flawed.'*

Response #3: We thank Reviewer #2 for mentioning this recent critic of the TMS-EEG method. In their article (Conde et al. 2019) and commentary on their paper (Siebner et al. 2019), Conde, Siebner et al., reported similarities in evoked responses produced by real and sham TMS when sham is applied in the most realistic conditions, because of non-transcranial multisensory co-stimulation. Nevertheless, we do not consider that these critics can be applied to our research, given the following points:

I) In their paper, Conde et al. state that *'The very early post-stimulation time bin (<20 ms after stimulation) was not considered to avoid the first strong TMS and electric stimulation related artefacts.'* Yet, in our study, the mean latency of the positive peak of this evoked responses is at 15.24 ± 2.97 ms (median = 15 ms) across subjects, which is made possible because our TMS-EEG apparatus is equipped with a proprietary sample-and-hold circuit, which guarantees TMS-

artefact free data ~5 ms post stimulation. (Virtanen et al., 1999, Med. Biol. Eng. Comput.). Therefore, for this reason alone, Conde et al. statements do not apply to our findings.

II) In Siebner et al. (2019), the authors state that *'It is possible that TMS-EEG settings using higher simulation intensities (>100 V/m, >130% of RMT) or larger coils may result in a more favourable ratio between on-target (transcranial) and off- target (peripheral) brain activation. We did not assess the impact of PEPs on TEPs at high TMS intensities (>100 V/m, >130% of RMT) or when using larger TMS coil diameters, which can be used to produce stronger and larger electric fields and thus a stronger synchronized response in larger neuronal populations.'* Mean electric field applied across all the TMS-EEG sessions in our sample was 115.48 ± 14.85 V/m, with only 5 subjects (~8% of our sample) with electric field lower than the mentioned 100 V/m. This concern may therefore only apply to a very limited portion of our sample.

III) In their paper, Conde et al. further report on similarities/differences between responses evoked following real TMS and realistic sham across different EEG channels (Figure 4, lower panel). At the earliest latencies used in Conde's paper (which is still late compared to our responses of interest), the median position of the electrodes we considered to compute cortical excitability in our sample falls inside the region where real TMS and sham evoked responses are not correlated (MNI space median position across the sample = [-34, -3.7, 85.9] mm). In other words, according to Conde et al. (2019), we measured cortical excitability over channels leading to measurable differences between real and sham TMS.

Overall, regarding these three points, it is evident that the issues raised by Reviewer #2 are based on methodological concerns that cannot be applied to our protocol and to our data so that they do not compromise at all the validity and interest of our results.

REVIEWERS' COMMENTS:

Reviewer #1 (Remarks to the Author):

The authors have improved the manuscript, e.g. with regard to the abstract and methodological details. The fact that the authors have considered individual chronotype and done thorough screening is also commendable. The responses to the issues raised by Reviewer #2 are also very thorough.

Comment: In their response, the authors write "(they were told during recruitment that the wake-extension protocol lasted for around 20 hours, until the middle of the night)". Please specify this in the main manuscript text to make this section even clearer.

Related to this comment: in the information provided in written form to the participants, were they not informed about the number of TMS sessions? (Would be important from an ethical standpoint, given that there is a risk of headaches and a very small risk of seizures that would still be important to consider when participants are considering the risk-benefit potential in enrolling)

Comment: "Unusual" physical exercise is very unspecific. This could mean that some participants still exercised just until the intervention day (e.g. doing intense daily exercise, because it was their "usual" exercise regimen/schedule), vs. others who just stuck to "usual" sedentary behavior and didn't exercise at all. If there is such variability, this can be specified as a caveat (if exercise/fitness levels weren't controlled for more specifically). Alternatively, it can perhaps be taken into account in the analysis (e.g. in a regression/correlation analysis with frontal CEP) to see if that influenced the results?

Reviewer #2 (Remarks to the Author):

The authors have addressed my concerns. Indeed the main aspects of the manuscript are much clearer and there is a considerable improvement avoiding misunderstandings of what the authors have done. In that light I do not have further problems with this manuscript.

Reviewer #3 (Remarks to the Author):

Van Egroo and colleagues present data from a cross-sectional study of 60 late middle-aged adults relating cognitive function to 1) the dynamics of cortical excitability assessed by TMS-EEG under conditions of prolonged wakefulness 2) NREM SWA 3) gray matter volumes 4) cortical amyloid and tau assessed by PET. They report that the temporal trend of cortical excitability in prolonged wakefulness (i.e. increasing vs. decreasing) is associated with cognitive performance (especially in executive function) independent of gray matter volumes, and independent of NREM SWA.

I have reviewed the appeal and rebuttal documents.

I agree that the finding of an association between the temporal dynamics of cortical excitability in prolonged wakefulness and cognitive performance in non-sleep-restricted conditions is a novel finding of interest to the field. However, I do note that interpretation of this association is not straightforward. The study design does not allow definitive differentiation of homeostatic (i.e. time awake) from circadian (i.e. time of day) processes which would require a study design that allowed dissipation of

homeostatic drive while preserving circadian effects (e.g. an ultradian constant routine protocol) or a study design that allowed exploration of all combinations of time awake and circadian phase (e.g. forced desynchrony), although the association with NREM SWA is somewhat in keeping with a homeostatic effect. Moreover, this cross-sectional observational study cannot distinguish between these three possibilities: 1) a direct causal role of wake/circadian modulation of cortical excitability on cognition, 2) wake/circadian function as a marker of an unmeasured neurodegenerative process that also contribute to cognition (e.g. reactive astrogliosis, microglial activation, microvascular dysfunction, etc), or 3) wake/circadian function as a marker of the integrity of subcortical circuits that contribute to both wake/circadian function and to cognition (and that weren't assessed by the imaging/EEG protocol in this study).

I agree that between their existing and new analyses the authors have provided reasonable evidence that the association between wake/circadian modulation of cortical excitability and executive function is independent of gray matter volumes / tau / amyloid in both 1) the area being stimulated and 2) areas involved in executive function more broadly.

I lack the expertise to assess the validity of the authors' arguments vis-a-vis the work of Conde el al. and Siebner et al.

Authors' Responses to Reviewers

Reviewer #1:

Comment #1: *'In their response, the authors write "(they were told during recruitment that the wake-extension protocol lasted for around 20 hours, until the middle of the night)". Please specify this in the main manuscript text to make this section even clearer.'*

Response #1: We added this information in the main text.

Page 21, lines 21-22: *'They were told during recruitment that the wake-extension protocol lasted for around 20h, until the middle of the night.'*

Comment #2: *'In the information provided in written form to the participants, were they not informed about the number of TMS sessions? (Would be important from an ethical standpoint, given that there is a risk of headaches and a very small risk of seizures that would still be important to consider when participants are considering the risk-benefit potential in enrolling)'*

Response #2: We added the following sentence in the main text to further clarify that point.

Page 22, lines 2-3: *'Participants were instructed that that the protocol included a few TMS-EEG session but were not informed of the exact number.'*

Comment #3: *'"Unusual" physical exercise is very unspecific. This could mean that some participants still exercised just until the intervention day (e.g. doing intense daily exercise, because it was their "usual" exercise regimen/schedule), vs. others who just stuck to "usual" sedentary behavior and didn't exercise at all. If there is such variability, this can be specified as a caveat (if exercise/fitness levels weren't controlled for more specifically). Alternatively, it can perhaps be taken into account in the analysis (e.g. in a regression/correlation analysis with frontal CEP) to see if that influenced the results?'*

Response #3: We modified the term 'Unusual' and we added *'exercise/fitness levels were not controlled for more specifically'*.

Page 20, lines 12-15: *'Aside from the fixed sleep-wake schedule, participants were also instructed to abstain from ~~unusual~~ intense physical exercise for the last 3 days of fixed-schedule circadian entrainment (i.e. right before the wake-extension protocol; exercise/fitness levels were not controlled for more specifically).'*

Reviewer #2:

No additional comments were added by Reviewer #2.

Reviewer #3:

We thank Reviewer #3 for recognizing the validity of the evidence presented in our paper, as well as for their constructive comments regarding the interpretation of the results.

Comment #1: *‘The study design does not allow definitive differentiation of homeostatic (i.e. time awake) from circadian (i.e. time of day) processes which would require a study design that allowed dissipation of homeostatic drive while preserving circadian effects (e.g. an ultradian constant routine protocol) or a study design that allowed exploration of all combinations of time awake and circadian phase (e.g. forced desynchrony), although the association with NREM SWA is somewhat in keeping with a homeostatic effect. Moreover, this cross-sectional observational study cannot distinguish between these three possibilities: 1) a direct causal role of wake/circadian modulation of cortical excitability on cognition, 2) wake/circadian function as a marker of an unmeasured neurodegenerative process that also contribute to cognition (e.g. reactive astrocytosis, microglial activation, microvascular dysfunction, etc), or 3) wake/circadian function as a marker of the integrity of subcortical circuits that contribute to both wake/circadian function and to cognition (and that weren't assessed by the imaging/EEG protocol in this study).’*

Response #1: We agree with the points raised by Reviewer #3. These considerations are already mentioned in the text at the following lines:

Page 15, lines 14-16: *‘While the contribution of the homeostatic process is most obvious in our data, a longer protocol covering the whole circadian cycle would help disentangle the respective modulation of frontal CEP by the circadian system.’*

Point 1):

Page 14, lines 15-17: *‘Furthermore, the predictive value of frontal CEP assessment for subsequent cognitive decline and risk of developing dementia remains to be investigated in a longitudinal protocol.’*

Page 15, lines 7-8: *‘Its cross sectional nature does not allow us to comment on the future cognitive trajectory of participants.’*

Point 2):

Page 15, lines 20-23: *‘Finally, we did not consider other age-related changes of brain integrity, such as cerebrovascular pathology, which are extremely common (up to 50%) as a mixed pathology in individuals with Alzheimer’s dementia, and Lewy bodies pathology, which shares some genetic risk with Alzheimer’s disease.’*

Point 3):

Page 14, lines 9-12: *‘It might also be the case that changes in cortical function stems from AD-related alteration of subcortical structures. In addition to the locus coeruleus, neurodegeneration of the suprachiasmatic nuclei, (SCN; site of the master circadian clock,) has been reported in AD while SCN network uncoupling in the suprachiasmatic nuclei is found in normal aging.’*